# Sperm Head Morphology Alterations Associated with Chromatin Instability and Lack of Protamine Abundance in Frozen-Thawed Sperm of Indonesian Local Bulls

**DOI:** 10.3390/ani13152433

**Published:** 2023-07-27

**Authors:** Asmarani Kusumawati, Faisal Amri Satrio, Rhesti Indriastuti, Zulfi Nur Amrina Rosyada, Berlin Pandapotan Pardede, Muhammad Agil, Bambang Purwantara

**Affiliations:** 1Department of Reproduction and Obstetric, Faculty of Veterinary Medicine, Universitas Gadjah Mada, Yogyakarta 55281, Indonesia; 2Division of Reproduction and Obstetrics, School of Veterinary Medicine and Biomedical Sciences, IPB University, Bogor 16680, Indonesia; faisalsatrio6@gmail.com (F.A.S.); rhinogil@apps.ipb.ac.id (M.A.); purwantara@apps.ipb.ac.id (B.P.); 3Tuah Sakato Technology and Resource Development Center, Department of Animal Husbandry and Animal Health of West Sumatra, Payakumbuh 26229, Indonesia; indriastutirhesti@gmail.com; 4Research Center for Applied Zoology, National Research and Innovation Agency (BRIN), Bogor 16911, Indonesia; zulfi.rosyada@gmail.com

**Keywords:** CMA_3_, local Indonesian bulls, protamine deficiency, PRM1 abundance, sperm DNA damage, sperm head defects

## Abstract

**Simple Summary:**

This study analyzed the association of various alterations in sperm morphology with the level of DNA damage, PRM deficiency, and the abundance of the PRM1 gene and protein in sperm. We found in this study that each parameter is associated with one another. Thus, gene expression and PRM1 protein abundance may play an essential role in the stability of sperm chromatin, especially concerning DNA damage and alterations in sperm head morphology. In addition, although it is not a novelty, the CMA3 in this study proved effective for bulls in Indonesia. It can be used as an alternative test to detect PRM content in sperm without having to carry out assessments at the molecular level, such as gene or protein expression assessments.

**Abstract:**

This study aimed to analyze various alterations in the morphology of the sperm head and its association with nucleus instability and insufficient sperm protamine. Frozen-thawed semen from twenty local Indonesian bulls was used for all stages in this study. The results of sperm head defect assessments are used for bull grouping, high (HD) and low (LD). Sperm DNA damage was assessed using Acridine Orange and Halomax. The PRM1 protein abundance was carried out using an enzyme immunoassay, while PRM1 gene expression was carried out using the RT-qPCR. PRM deficiency was performed using CMA_3_. Several kinds of sperm head defects in the HD were significantly higher (*p* < 0.05) than in the LD bulls. Sperm DNA damage showed a significant (*p* < 0.05) difference between the HD and LD bulls. PRM1 abundance was significantly (*p* < 0.05) decreased in HD bulls. PRM deficiency was significantly (*p* < 0.05) higher in HD bulls than in LD bulls. PRM deficiency in bulls correlated significantly (*p* < 0.01) with sperm head defects, DNA damage, and PRM1 abundance. The lack of sperm protamine might affect the sperm nucleus’s stability and induce morphological alterations in the sperm head.

## 1. Introduction

Increasing the local cattle population is the goal of one of the Indonesian government’s programs to reduce dependence on imported beef and feeder cattle. The improvement program highlights the various advantages of local cattle, including the excellent adaptability of local cattle to tropical climates, a high level of resistance to various animal diseases, and increased productivity capabilities in extensive rearing with relatively limited feed [1]. Also, local cattle generally have fairly efficient reproductive performance, with relatively fast body weight growth, and the taste of meat from local cattle is highly favored by some residents [2]. This improvement program can be realized through an artificial insemination (AI) program using frozen semen. Good-quality frozen semen must contain normal and mature sperm capable of carrying and transmitting intact DNA, contributing to the genetic information essential for fertilization [3].

In spermatogenesis, the typical structure of normal and mature sperm is formed through a series of complex cellular processes. If various changes occur during the process, they will impact morphological defects. A sperm head abnormality is a major abnormality in sperm morphology, and a high percentage will negatively affect fertility [4]. Pardede et al. [3] reported that bulls with total sperm abnormalities (head and tail defects) of 3–5% had an in vivo conception rate of up to 79–84%. In comparison, the percentage of total sperm abnormalities is >10%, and the conception rate only reaches <66%. Sperm head abnormalities are mainly caused by the disruption of the spermatogenesis process, such as the disruption of the stages of meiosis, impaired maturation during transit in the epididymis, disturbances in cell nucleus condensation, testicular degeneration, and disorders of ectoplasmic specialization in Sertoli cells and acroplasomes, which are involved in the formation of the cell nucleus [4]. Ball and Peters [5] reported that the morphological defects in the sperm head were related to genetic factors, chromosomal abnormalities, impaired sperm nuclei chromatin condensation, and DNA damage. DNA damage in sperm will impact the failure of the embryogenesis process and is one of the factors that cause low fertility [6]. Various effects of DNA damage on sperm have been widely reported, such as an increase in the incidence of abortion, a reduction in fertility rate, poor embryonic development, and lower pregnancy rates [6]. Boe-Hansen et al. [7] reported early embryonic mortality and offspring size due to DNA damage in sperm. Sperm DNA damage can be influenced by various factors, including environmental factors, genetic factors, disruption during spermiogenesis, and abnormalities in chromatin structure [8]. Furthermore, protamine (PRM) aberrations have been linked to sperm DNA damage [6].

The primary protein in the sperm nucleus, PRM, is in charge of packing sperm chromatin. In spermiogenesis, mammalian spermatids undergo various changes, including the occurrence of the chromatin condensation process [9]. The chromatin condensation process is initiated by PRM, rich in arginine and cysteine residues [8]. The arginine content makes the PRM bond with DNA very strong. At the same time, the cysteine residue facilitates the formation of disulfide bonds between PRM with one and the intra-PRM disulfide bonds to pack chromatin optimally to support normal sperm function. As a result, PRM organizes sperm DNA optimally to promote chromatin condensation, which preserves the parental genome’s genetic integrity from nuclease enzymes, mutagens, and other stimuli that can damage DNA [8]. PRM is divided into PRM1 and PRM2 [8], even though PRM3 has been reported in the past [10]. PRM1 is the most common PRM variant, and it is essential to the proper functioning of bovine sperm [8]. Decreased expression or abnormal PRM content has been shown to influence DNA damage and fertility in males. Chromomycin A_3_ (CMA_3_) staining has been developed and reported to identify anomalous PRM content [11].

CMA_3_ is a specific fluorochrome guanine-cytosine that competes with the PRM binding site on DNA. Therefore, the low PRM content binds chromomycin to DNA fluorescently coloring sperm [11]. The association of PRM deficiency testing using CMA_3_ with sperm DNA damage has been widely reported and has been widely used [11]. As a result, CMA_3_ is a robust and well-established method for evaluating sperm chromatin packing, allowing for the indirect observation of sperm with aberrant or inadequate PRM content. Thus, this study is expected to be able to prove and further strengthen this hypothesis through the analysis of various alterations in the morphology of the sperm head and its association with nucleus instability and low sperm protamine. Proving the credibility of the CMA3 test in this study is expected to enrich the hypothesis regarding the superiority and effectiveness of this test as a test for semen quality, especially for assessing protamine content, DNA damage, and sperm head defects. In particular, this study presents the latest findings related to these issues in local Indonesian bulls’ semen that have not been previously reported.

## 2. Materials and Methods

### 2.1. Experimental Animals and Design

This study used commercial frozen semen and did not involve animals directly. All procedures in semen processing until commercialization follows the operational standards in Singosari AI Center, East Java, Indonesia, namely SNI ISO 9001: 2015 No. G.01-ID0139-VIII-2019 and supervised by a fully responsible veterinarian considering every aspect of animal welfare in Indonesia. The research was conducted in early 2022 during the rainy season in Indonesia. The Singosari AI Center, East Java, provided commercial frozen semen for this study. The frozen semen used was frozen semen from local Indonesian cattle used for the Artificial Insemination program in Indonesia. Twenty Indonesian local bull bulls with an age range of 4 to 5 years, in healthy and productive condition, were used in this study. All bulls were in the same location, with the same type of feed, a body weight of around 700 kg, and an environment with the same average temperature (20–22 °C). Feed was given to the bulls in the form of feed that has been mixed using the TMR (Total Mixed Ratio) method. Complete feed, or TMR, consists of forages and concentrates in balanced and adequate conditions. The feed ingredients included elephant grass, concentrate, silage, hay, and minerals. Fresh semen with more than 70% sperm motility, 800 × 10^6^/mL sperm concentration, and <20% sperm abnormalities was cryopreserved. The frozen semen used in the study had undergone a post-thawing motility check process using computer-assisted sperm analysis (CASA) and met the distribution requirements, progressive motility ≥ 40%.

All bulls used in the study were classified into two groups based on the level of the morphological defects of the sperm head. Evaluation of sperm head morphological defects was carried out using carbol-fuchsin and eosin staining based on Pardede et al. [3]. Two hundred frozen semen straws (ten straws/bull) from twenty local Indonesian bulls were used in this stage. Frozen semen straws were thawed in a water bath at 37 °C for 30 s before being placed in a microtube. The semen samples smeared on glass slides were then fixed with Bunsen and absolute alcohol. After drying, the sample slide was rinsed in a 2% chloramine solution for ±2 min and washed with distilled water and 95% alcohol. Next, the sample slide was stained with carbol-fuchsin and eosin staining for six minutes, washed in running water, and dried. Observations were made with a light microscope with a magnification of 400×. Sperm head morphological defects were counted from a total of 500 sperm cells. Twelve variants of sperm head defects were found and observed in this study, namely abaxial, knobbed acrosome, abnormal contour, macro-micro cephalic, pear shape, undeveloped, round head, detached head, double head, narrow at the base, and tapered (Figure 1). Sperm head defects from each bull used in the study were calculated as the average. The average value of sperm head defects was 2.96%, ranging from 1.88% to 4.10%. Based on the average value of sperm head defects and the differences from the population average, the bulls were then categorized as high sperm head morphological defects (HD) and low sperm head morphological defects (LD). Our cutoff for driving a wedge into HD and LD categories is the mean of the entire population. Bulls with higher-than-average values of sperm head defects were classified as HD, whereas those with low sperm head defects were classified as LD. Table 1 summarizes the classification of bulls used in this work based on sperm head defects. These grouping results are then used to cluster further assessments, such as DNA damage, PRM deficiency, and PRM1 protein and gene abundance.

### 2.2. Frozen-Thawed Sperm Analysis

#### 2.2.1. DNA Damage Assessment by Acridine Orange Staining

Evaluation of DNA damage with acridine orange staining was carried out based on Pardede et al. [12]. Semen from previous sperm head morphological evaluation (200 frozen semen straws (10 straws/bull) from 20 local Indonesian bulls) was used in this stage. The semen samples were first smeared on glass slides and dried in the air. The sample slide was then fixed for two hours in Carnoy’s solution (methanol/glacial acetic acid) and air-dried—and, once dry, stained with freshly prepared acridine orange stain for five minutes in the dark. Then, the sample slide was washed with distilled water. Five hundred sperm cells were observed and counted under a fluorescent microscope at 450–490 nm wavelength. Sperm with normal DNA fluoresced green, but sperm with fragmented DNA fluoresced in a yellow-green to orange spectrum (Figure 2).

#### 2.2.2. DNA Damage Assessment with Sperm-Bos-Halomax

The study of DNA damage with the Sperm-Bos-Halomax was conducted based on Pardede et al. [13]. Another frozen thawed-semen straw was used in this stage. Two hundred frozen semen straws (ten straws/bull) from twenty local Indonesian bulls were used. A total of 50 μL of liquid agarose was put in a tube containing 25 μL of the semen sample (15–20 × 106 cells/mL) at 37 °C and mixed carefully. First, we placed a drop of 1.5–2 μL of cell suspension on the marked wells of the pre-cooled glass plate (4 °C), covered the four wells with a 24 × 24 mm coverslip, and transferred them into the fridge at 4 °C for 5 min. Then, we removed the slides from the refrigerator, carefully removed the coverslip at room temperature (22 °C), and placed them horizontally over the float in a Petri dish. Next, we dried the lysis solution on the slides for 5 min. Next, we washed the slides for 5 min with distilled water and dried them by tilting the slides. Then, the slides were dehydrated in ethanol solution (70% and 90%) for 2 min each and left to dry. In the final step, we stained the slides with 5 μL of 0.01 mm propidium iodide for 5 min. A total of 500 sperm cells were observed under a fluorescence microscope. Sperm with intact DNA has a small halo, whereas sperm with fragmented DNA has a wide halo (Figure 2).

### 2.3. Measurement of Bovine Sperm Protamine 1 Protein and Gene Abundance

PRM1 protein concentration was measured using enzyme immunoassay (EIA). Another frozen thawed-semen straw was used in this stage. Two hundred frozen semen straws (ten straws/bull) from twenty local Indonesian bulls were used. A 100 µL semen sample was centrifuged for 15 min and washed twice with a phosphate-buffered saline solution (PBS). The precipitated sperm was then used for further EIA. Sperm samples and standard Bovine PRM1 samples were inserted into suitable wells. We covered the reaction wells with adhesive tapes, hatching them in an incubator at 37 °C for 90 min. After that, we washed the plates twice and filled each well with 100 μL of the biotinylated antibody working solution. We covered and hatched them. Then, we returned them to the incubator for 60 min at 37 °C. We rewashed the plate three times, then covered and hatched for 30 min at 37 °C with 100 μL of the enzyme working solution. We then rewashed the plate five times, added 100 μL of the color reagent solution, and hatched for 30 min at 37 °C in a dark incubator. Then, we carefully mixed 100 μL of color reagent C. Using an EIA reader with a wavelength of 450 nm, the concentration of Bovine PRM1 was determined [13].

According to Pardede et al. [13], the PRM1 gene expression level was evaluated using the RT-qPCR technique. At this point, another frozen thawed-semen straw was used. Four hundred frozen semen straws (twenty straws/bull) from twenty local Indonesian bulls were used in this stage. Each frozen semen straw was defrosted in a water bath, washed three times with phosphate-buffered saline (PBS), and centrifuged at 16,000× *g* for 15 min to pellet the sperm (approximately 25 × 10^6^ sperm cells/mL). The TRI reagent was then used to extract the total RNA as directed by the manufacturer (Zymo Research, Irvine, CA, USA). The quantity and purity of the total RNA were quantified using the NanoDrop^TM^ One/OneC Microvolume UV-Vis Spectrophotometer (Thermo Scientific, Marsiling Industrial Estate Rd 3, Singapore). The SensiFAST^TM^ cDNA Synthesis Kit (Bioline Ltd., Memphis, TN, USA) was also utilized for cDNA synthesis, following the manufacturer’s guidelines. This procedure produced 20 μL of cDNA that was prepared for RT-qPCR. The quantity of the transcripts was also determined using the quantitative real-time PCR (qPCR), and the reactions were carried out using the SsoFast^TM^ EvaGreen Supermix (Bio-Rad Lab, Hercules, CA, USA). The study’s assessed genes included PPIA (XM_001252921.1; forward: ATGCTGGCCCCAACAA-3′ and reverse: 5′-CCCTCTTTCACCTTGCCAAA-3′) as a housekeeping gene and PRM1 (BC108207; forward: AGATACCGATGCTCCTCACC-3′ and reverse: 5′-GCAGCACACTCTCCTCCTG-3′). The PRM1 gene’s relative levels of expression were compared to those of the housekeeping gene PPIA using the 2^−∆∆CT^ method.

### 2.4. Protamine Deficiency Assessment with CMA_3_ Staining

The CMA_3_ staining method developed by Carreira et al. [11] was used to assess sperm PRM deficiency. Another frozen thawed-semen straw was used in this stage. Two hundred frozen semen straws (ten straws/bull) from twenty local Indonesian bulls were used. The semen samples were smeared on glass slides and fixed in a methanol/glacial acetic acid (3:1) solution for 5 min at 4 °C. Then, for 20 min, we soaked the sample slides in a CMA_3_ solution (0.25 mg/mL CMA_3_ in McIlvane’s buffer (pH 7.0) supplemented with 10 mM MgCl_2_). The slides were then rinsed in McIlvane’s buffer and dried as directed. A fluorescent microscope was used to examine 500 sperm cells. Positive CMA_3_ or PRM deficiency in sperm was indicated by bright green fluorescence on the sperm head. In contrast, negative CMA_3_ or sperm with intact PRM was indicated by faint green staining on the sperm head.

### 2.5. Statistical Analysis

Twenty bulls (*n* = 10 HD, *n* = 10 LD) were utilized for the statistical test. The data with a sperm head defects trait were analyzed using a generalized linear mixed model. Data analysis to determine the grouping of breeding bulls based on the f sperm head defects phenotype has been explained in the previous subsection. The normality test on the research data was carried out using the Shapiro–Wilk test, which was then tested for homogeneity using the Levene test. The research data are normally distributed and vary homogeneously. The statistics were analyzed using SPSS version 25.0 (IBM, Armonk, NY, USA) and the student T-test methodology. In addition, the correlation between CMA_3_ and various semen parameters in this study was analyzed using Pearson correlation. The data are presented in tables and graphical diagrams with the mean ± standard error of the mean (SEM).

## 3. Results

### 3.1. Characteristic Sperm Head Morphological Defects in HD and LD Bulls

Overall, sperm head defects in the HD group were significantly higher (*p* < 0.05) than those in the LD group (4.03 ± 0.13% vs. 1.91 ± 0.13%). There was a significant difference between HD and LD bulls (*p* < 0.05) in pear-shape defects (0.29 ± 0.03% vs. 0.12 ± 0.03%), tapered heads (0.29 ± 0.03% vs. 0.19 ± 0.02%), abnormal contour (0.93 ± 0.05% vs. 0.41 ± 0.03%), macrocephalic (0.75 ± 0.05% vs. 0.24 ± 0.02%), microcephalic (0.59 ± 0.06% vs. 0.12 ± 0.02%), undeveloped (0.19 ± 0.01% vs. 0.07 ± 0.03%), narrow at the base (0.17 ± 0.03% vs. 0.05 ± 0.02%), and double heads (0.07 ± 0.03% vs. 0.00 ± 0.00%). There was no difference in sperm with a round head, knobbed acrosome, detached head, and abaxial defects in HD and LD bulls (Figure 3).

### 3.2. Sperm DNA, PRM Deficiency, and PRM1 Abundance in HD and LD Bulls

Sperm DNA damage using AO staining (2.78 ± 0.14% vs. 2.10 ± 0.05%) and the Sperm-Bos-Halomax (3.38 ± 0.07% vs. 2.20 ± 0.05%) showed a significant difference (*p* < 0.05) between the HD and LD bull’s groups (Figure 4). An assessment of PRM deficiency with CMA_3_ showed that sperm PRM deficiency was higher in HD than in LD bulls (2.68 ± 0.09% vs. 1.90 ± 0.05%) (Figure 4). A PRM1 protein abundance assessment using EIA showed a significantly decreased PRM1 protein abundance in HD bulls (*p* < 0.05) (0.170 ± 0.01 ng/mL vs. 0.243 ± 0.01 ng/mL) (Figure 4) compared to LD bulls. A PRM1 gene abundance assessment using RT-qPCR showed a significant difference (*p* < 0.05) (4.64 ± 0.23 vs. 1.68 ± 0.41) between the HD and LD bull’s groups.

### 3.3. Correlation of PRM Deficiency, Sperm Head Defects, DNA Damage, and PRM1 Abundance

Sperm PRM deficiency in bulls had a high and significant correlation with sperm head defects (*p* < 0.000), DNA damage by AO staining (*p* < 0.000), and the Sperm-Bos-Halomax (*p* < 0.000), PRM1 protein abundance (*p* < 0.003), and PRM1 gene expression (*p* < 0.001). Sperm head defects in bulls were highly and significantly correlated with AO staining (*p* < 0.000), Sperm-Bos-Halomax (*p* < 0.000), PRM1 protein abundance (*p* < 0.000), and PRM1 gene expression (*p* < 0.001). DNA damage by AO staining in bulls was significantly correlated with the Sperm-Bos-Halomax (*p* < 0.000), PRM1 protein abundance (*p* < 0.002), and PRM1 gene expression (*p* < 0.001). The results of the Sperm-Bos-Halomax in bulls also have a high and significant correlation with the PRM1 protein abundance (*p* < 0.001) and PRM1 gene expression (*p* < 0.002) (Table 2).

## 4. Discussion

This study analyzed the structure of sperm chromatin to determine the relationship between sperm head morphology alterations with DNA instability and sperm PRM content in local Indonesian bulls. Twelve variants of sperm head defects were observed in this study. These alterations in sperm morphology were significantly positively correlated with DNA damage either by AO staining or by Sperm-Bos-Halomax^®^. Abnormal contour, undeveloped, macro-micro cephalic, tapered, narrow at the base, double head, and round head defects are associated with genetic factors, chromosomal abnormalities, DNA damage, and sperm chromatin disorders [14,15,16]. Abnormal chromatin condensation triggers the elasticity of the sperm cell nucleus, resulting in an irregularity in the form of the sperm head that affects the sperm head with an abnormal contour [17]. The undeveloped defect is a defect in sperm in which the sperm does not develop. It can be small, and further examination found that sperm cells with this defect are not composed of complete genetic material [16]. In sperm, microcephalus abnormalities are linked to problems with cell nucleus condensation [18]. In contrast, macrocephalus defects can be caused by genetic factors such as DNA or chromosome abnormalities [17]. The tapering defect in sperm is linked to a breakdown of ectoplasmic specialization in Sertoli and archoplasms cells involved in cell nuclei production [19].

Narrow at the base defects are linked to sperm DNA damage [20]. A disruption in epididymis transit has also been identified as a cause of this condition [21]. Double head defects in sperm occur due to interference with the cytogenesis process or the formation of intracellular bridges during meiotic division, which causes imperfect separation of the sperm [22]. This defect has also been linked to DNA damage [20]. Other sperm head defects reported to have a genetic influence include round-head defects, pear-shaped defects, and knobbed acrosome. A round head defect is a defect in sperm characterized by a round head of the sperm without an acrosome boundary associated with genetic disorders [16]. Pear-shaped defects can be caused by abnormal conditions resulting from changes in testicular function, such as testicular temperature regulation or hormonal disturbances [16]. They can be passed on to offspring [16]. Donald and Hancock [23] reported that a high-knobbed acrosome defect is closely related to autosomal recessive sex. A delay in forming the acrosomal phase at the time of spermiogenesis is the cause of this defect [16]. Apart from genetic factors, environmental factors, such as hormonal imbalances, stress, nutritional deficiencies, and extreme climatic conditions, can also influence sperm abnormalities [24].

Overall, each defect in the sperm head in this study did not exceed 1%, although there was a significant difference (*p* < 0.05) between the HD and LD bulls’ groups in several variants of sperm head defects. Chenoweth [16] revealed that bulls with good fertility have at least a proportion of less than 1% in each variation of sperm head abnormalities. Abnormal contour, undeveloped, pear-shaped, macro-micro cephalic, and knobbed acrosome defects in high numbers can cause sperm not to perform its fertilization function and reduce fertility [17]. Detached head and abaxial defects were found in small numbers in this study. The detached head is characterized by separating the head from the sperm tail, which occurs in the epididymis [25]. Chenoweth [16] reported that bulls that were in a sexually inactive status for several weeks experienced a rise in the proportion of detached heads. The abaxial defects found in this study were characterized by the position of the sperm tail, which was not in the middle, and were reported not to affect fertility [16].

However, the various types of alterations in the morphology of the sperm head must be taken into account, mainly if they are observed in large numbers, as they might interfere with male fertility. Bulls with overall morphological defects >17%, according to Ball and Peters [5], do not have good reproductive efficiency. According to Patel et al. [26], the overall morphological defect in bulls was 20%, with general morphological defects ranging from around 8–12 percent in bulls with good fertility. The sperm head defects in this study were classified as usual, namely less than 5%. Pardede et al. [3] reported that sperm with a percentage of total sperm abnormalities of 3–5% have a conception rate of up to 79–84%. The percentage of total sperm abnormalities was >10%; the conception rate only reached <66%. In their study, Petac and Kosec [27] reported that the overall conception rate in bulls with sperm head defects of 4.0% was 73.3%. Another study reported that a bull with 36% sperm head defects resulted in a pregnancy rate of 46% [28]. Anderson et al. [29] also reported that sperm head defects, especially those related to abnormalities in the acrosome area in bovine sperm, with a 23–28% percentage, resulted in an average 60-day nonreturn rate of 48.8–62.3%. Although this study has not analyzed the fertility rate of each bull used, referring to the findings and based on previous studies, perhaps the sperm head defects in this study, which are only less than 5%, can achieve a pretty good percentage conception rate. Even that can be achieved if other factors besides the bull factor in the artificial insemination program can be controlled to achieve optimal fertilization rates.

The stability of sperm DNA in chromatin is one factor that must be maintained, given its essential role in successful fertilization and embryonic growth [3]. This study also found that DNA instability resulted in various alterations in the sperm head. Increased DNA damage was found in bulls in the group with high sperm head abnormality in the bulls. DNA damage assessment was carried out in this study using AO staining and the Sperm-Bos-Halomax^®^. Evenson [30] reported that the SCSA (sperm chromatin structure assay) is the most stable test for DNA damage compared to other tests, and AO staining is the most insensitive test compared to the others. Halomax is the most straightforward chromatin DNA test because it is included in a KIT combined with various cell dyes and can be used with light or fluorescence microscopy [30]. However, the sperm chromatin packaging instability will cause DNA damage and disperse, forming a large halo. Both were still usable in this study and significantly correlated with PRM deficiency and sperm head abnormality. DNA damage that occurred in this study as a whole was less than 4%. Previous research reported that DNA damage in sperm that is less than 15% is still expected, whereas 15% to 25% decreases fertility, and sperm with DNA damage more significant than 25% are categorized as sterile [31]. Pardede et al. [31] reported that DNA damage could reach 3–4% in bulls at productive age and still produce a fertility index from in vivo fertilization up to >70%. Thus, DNA damage in this study is still classified as normal, and the bulls have good potential regarding fertility traits. Overall, the formation of sperm morphology, or what is known as the process of sperm morphogenesis, plays an important role in ensuring the normal function of sperm. Various unique and complex mechanisms, including molecular mechanisms, are important keys to the morphological integrity of sperm. The failure of such morphogenesis can lead not only to structural defects of sperm and reduced sperm maturity but also to the fragmentation of sperm nuclear DNA due to impaired repair of damaged nuclear DNA strands during early spermatid stages. Sperm with reduced maturity will produce higher levels of reactive oxygen species and increased DNA damage. Finally, oxidative stress, ‘abortive’ apoptosis of differentiated germ cells, and teratozoospermia can arise. Furthermore, of the molecular mechanisms involved in the morphogenesis process, one is the mechanism of action of protamine, which is the main protein of the sperm nucleus and is involved in the condensation of spermatid chromatin.

The high PRM abundance in sperm influences DNA stability and the shape of the sperm head [3,8]. In our study, PRM abundance was significantly decreased in the bull group with high head defects. This was followed by significant DNA damage, both by AO staining and the Sperm-Bos-Halomax^®^ in this study. Moreover, this study demonstrated a close correlation between PRM, DNA instability, and changes in sperm head morphology. Zhu et al. [4] stated that head defects occur due to abnormal spermiogenesis. A sperm chromatin modification process happens at this phase, regulated by histone turnover via PRM and transitional proteins [9]. PRM will replace up to 85 percent of the histone sperm nucleus during this period [8]. PRM, which is the main protein in sperm, has a role in binding DNA, is the primary controller of the presence or absence of DNA damage, and includes changes in sperm head morphology. The disruption of PRM will trigger various alterations, including impaired chromatin compaction that can induce DNA breaks, which can disrupt the DNA backbone and indirectly appears to be the major cause of poor semen morphology. However, more in-depth studies are still needed to enrich the findings in this study, one of which is adding fertility rate data and analyzing its association with various sperm head defects found in this study. In this way, the influence of sperm head defects due to DNA instability and a lack of protamine abundance on the biological aspects of bulls, especially fertility, will be mapped more clearly.

CMA_3_ staining was significantly inversely associated with high amounts of PRM abundance in sperm in this study. This means that the CMA_3_ staining has been proven to be an alternative test for PRM content in sperm. However, this study strengthens and enriches previous studies regarding using CMA_3_ to assess chromatin packaging with abnormal or deficient PRM conditions [11]. Furthermore, CMA_’_’s fluorochrome-specific guanine-cytosine competes with DNA’s PRM binding site. Therefore, chromatin packaging with PRM deficiency conditions will impact the bonding between CMA_3_ staining and DNA [11]. In sperm with DNA chromatin packaging, which is compacted and stabilized by PRM, arginine will inhibit CMA_3_ staining [32].

Furthermore, the close correlation between the CMA_3_ protamine deficiency tester and DNA fragmentation, as well as sperm head defects, further suggests the possibility of CMA_3_ being a useful tool that can simultaneously assess all three parameters at once, PRM1 content, DNA fragmentation, and sperm head defects. Unfortunately, this study has not been able to provide fertility data to enrich this study and increase the usefulness of this test. As reported by Nasr-Esfahani et al. [33], CMA_3_ staining as an indicator of protamine content, along with semen analysis, could be a useful full test for evaluating fertility status in sub-fertile cases. Although recently fertility assessments have been developed based on sperm genes and protein [34,35], considering that the CMA_3_ test has good sensitivity and specificity values and is much easier and cheaper, there is nothing wrong with this test being the right choice for testing bull fertility.

## 5. Conclusions

This study proves and strengthens the hypothesis regarding the association between alterations in sperm head morphology and DNA damage, PRM deficiency, and the abundance of the PRM1 gene and protein. There were twelve types of sperm head defects found in this study, some of which were higher in the HD bull group. Sperm DNA damage and protamine deficiency were higher in the HD bull group, and conversely, the abundance of the PRM1 gene and protein was lower in the HD bull group. There is a close association between sperm head defects, protamine deficiency, and the abundance of the PRM1 gene and protein. In addition, although it is not a novelty, the CMA_3_ in this study proved effective for bulls in Indonesia. It can be used as an alternative test to detect PRM content in sperm without having to carry out tests at the molecular level, such as gene or protein expression tests. Furthermore, considering that CMA_3_ is much cheaper than testing at the molecular level, the use of CMA_3_ might also be very useful to apply to several developing countries whose economic systems are less stable.

## Figures and Tables

**Figure 1 animals-13-02433-f001:**
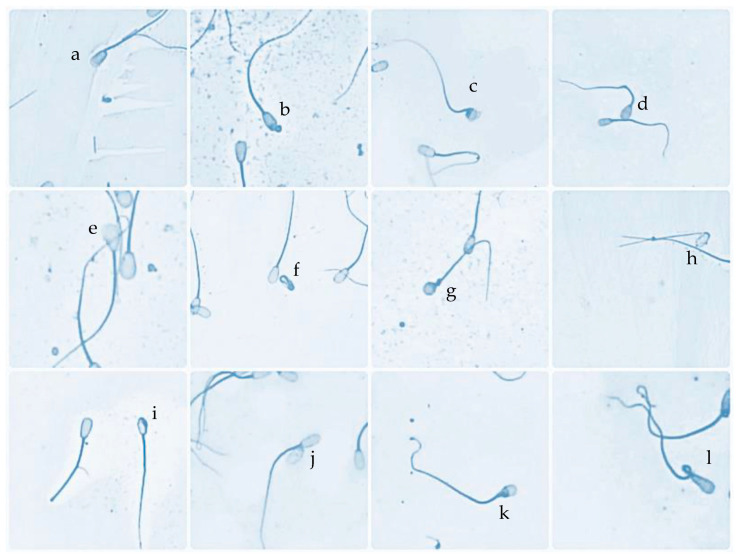
Sperm head morphological defects were found in the bulls: abaxial (**a**), knobbed acrosome (**b**), abnormal contour (**c**), normal and macrocephalic (**d**), pear shape (**e**), undeveloped (**f**), round head (**g**), detached head (**h**), microcephalic (**i**), double head (**j**), narrow at the base (**k**), and tapered (**l**).

**Figure 2 animals-13-02433-f002:**
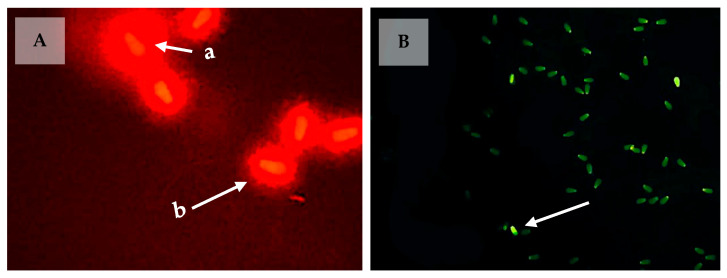
Sperm DNA damage using the Sperm-Bos-Halomax (**A**) method and PRM deficiency with CMA_3_ staining (**B**). The sperm with fragmented DNA showed a large halo as a marker of dispersed DNA chromatin (a); normal sperm DNA showed a slight halo as a marker of the density of sperm chromatin nucleus (b). The bright green sperm (CMA_3_+) indicates PRM deficiency (arrow), while the dull green sperm (CMA_3_−) indicates normal content.

**Figure 3 animals-13-02433-f003:**
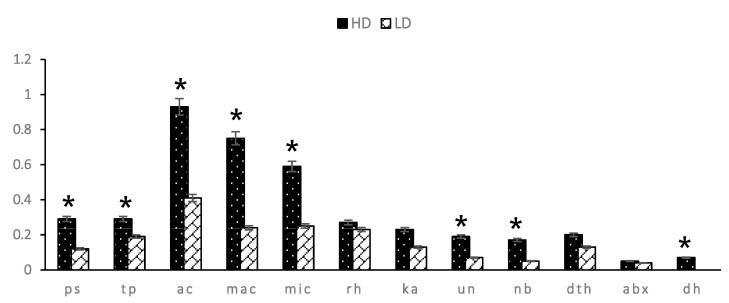
Graph the value distribution of various sperm head defects in HD and LD bulls. Sperm head defects sequentially include pear-shaped (ps), tapered (tp), abnormal contour (ac), macrocephalic (mac), microcephalic (mic), round head (rh), knobbed acrosome (ka), undeveloped (un), narrow at the base (nb), detached head (dth), abaxial (abx) and double head (dh). * Significant difference when compared to LD bulls (*p* < 0.05).

**Figure 4 animals-13-02433-f004:**
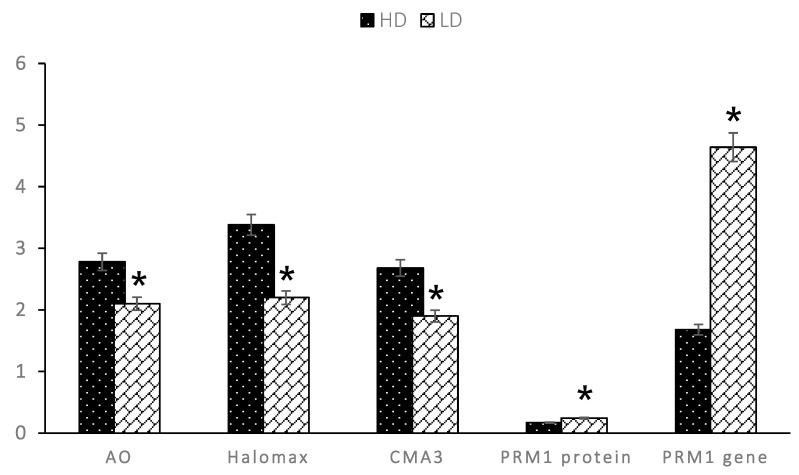
Graph the value distribution of DNA damage by AO staining and Sperm-Bos-Halomax, PRM deficiency by CMA_3_ staining, and PRM1 by EIA in HD and LD bulls. * Significant difference when compared to HD bulls (*p* < 0.05).

**Table 1 animals-13-02433-t001:** Sperm head morphological defects phenotypes of the bulls used for further analysis: Bulls 1–10 were defined as high sperm head morphological defects (HD), and Bulls 11–20 were grouped as low sperm head morphological defects (LD).

Bull No.	Sperm Head Morphological Defects Status	Average Sperm Head Morphological Defects (%)	Difference from Population Average (%)
1	High sperm head morphological defects (HD)	4.04	1.076
2		4.02	1.056
3		4.10	1.136
4		4.06	1.096
5		3.96	0.996
6		4.00	1.036
7		4.00	1.036
8		4.06	1.096
9		4.02	1.056
10		4.08	1.116
11	Low sperm head morphological defects (LD)	1.90	−1.064
12		1.92	−1.044
13		1.88	−1.084
14		1.92	−1.044
15		1.92	−1.044
16		1.92	−1.044
17		1.92	−1.004
18		1.96	−1.004
19		1.92	−1.044
20		1.88	−1.084
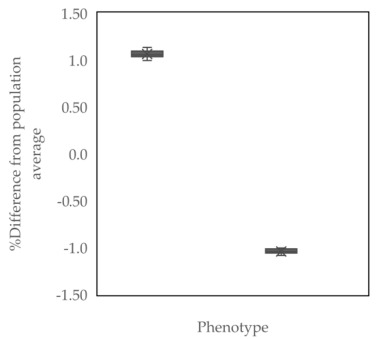			

Bulls were classified as having high sperm head morphological defects (HD) and low sperm head morphological defects (LD) based on average sperm head morphological defects scores and the percent differences from the population average (*p* < 0.05).

**Table 2 animals-13-02433-t002:** Correlation data of PRM deficiency, sperm head defect, DNA damage (AO staining and Sperm-Bos-Halomax), and PRM1 abundance in the bulls, regardless of bull grouping based on sperm head defects.

Sperm Parameters	PRM Def	Head Defects	AO	Bos-Halomax	PRM1-p	PRM1-g
PRM def	1	0.964 **	0.982 **	0.995 **	−0.936 **	−0.906 **
Head defects		1	0.954 **	0.953 **	−0.952 **	−0.912 **
AO			1	0.991 **	−0.926 **	−0.917 **
Bos-Halomax				1	−0.916 **	−0.921 **
PRM1-p					1	0.932 **
PRM1-g						1

PRM def: Protamine deficiency; PRM1-p: protamine-1 protein abundance; PRM1-g: protamine-1 gene expression. ** Correlation is significant at the 0.01 level.

## Data Availability

Not applicable.

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
