# Peer review of "Sperm Head Morphology Alterations Associated with Chromatin Instability and Lack of Protamine Abundance in Frozen-Thawed Sperm of Indonesian Local Bulls"

_animals, 2023, doi:10.3390/ani13152433_

Round 1

Reviewer 1 Report

The present research provides interesting information. However, some important changes need to be made before final publication.

Abstract: review the "Journal" guidelines. It is mentioned in "MDPI Style Guide" the following: "The abstract contains a summary of the entire article and can be up to 200 words with a single paragraph." (https://www.mdpi.com/authors/layout) In this case it exceeds the number of words. Therefore, restructure this section.

INTRODUCTION

General comments: I recommend adding more information on possible causes of sperm head abnormalities.

Line 63.- the authors mention “Pardede et al. [3] reported that sperm with a portion of total sperm abnormalities of 3-5%” estas anormalidades solo corresponden a “sperm head abnormality” ó generals?. I recommend mentioning what these anomalies would be.

Line 64.- change the word "in vivo" to "in vivo" in italics

MATERIALS AND METHODS

General comments: I recommend mentioning which were the selection criteria for these animals, body condition, etc. Also, if a complete clinical study was performed before the study to rule out any pathology.  On the other hand, why only the frozen semen was evaluated and the semen before freezing was not considered in order to observe the pre and post freezing effect? I recommend mentioning the number of samples evaluated for each animal.

Line 130-131.- the authors mention "Sperm head defects in the HD group were significantly higher (P<0.05) than those in the LD group (4.03±0.13% vs. 1.91±0.13%)." I recommend moving this sentence to the "Result" section under "Characteristic sperm head morphological defects in HD and LD bulls". In addition it would be convenient to mention that sperm with head defects were considered within the "HD" and "LD" groups as the case may be.

Line 113.- in the "Statistical analysis" section specify if the data came from a normal population (what tests were performed?). If they were not normal data, did you use transformed data? Also, why did you use a "Pearson correlation" and did not perform a "Sperman correlation" with the variables evaluated? Also, mention what level of significance was considered (P<0.05 or P<0.01). I suggest describing this section further.

RESULTS

General comments:

I recommend restructuring the figures and using the same monochromatic format since fig 3 is represented differently from fig 4 for the "HD" and "LD" groups. As well as the labels of each axis.

Line 235-236.- the authors mention "There was no difference between round head (rh), knobby acrosome (ka), detached head (dth), and abaxial (abx) in HD and LD bulls." This was already mentioned in line 228. Line 238. Below the figures is only to mention and describe the labels of the data. I recommend not to repeat it.

Line 262.- Figure 4, I recommend moving it to line 247, below the section "Sperm DNA, PRM deficiency, and PRM1 abundance in HD and LD bulls". Because there is a junction with "Table 1".

Line 264.- Figure 4, it mentions "DNA damage and PRM deficiency in sperm were significantly more important in HD bulls than in LD bulls (P <0.05). In addition, PRM1 concentration was significantly higher in LD bulls than in HD bulls (P <0.05)." This should go in the text not in the figure. The text in the figure is only to describe abbreviations.

DISCUSSION

In general, I recommend an explanation of the findings observed in the research and try to explain the mechanism involved. I recommend restructuring this section because the first part of the discussion focuses mainly on morphological alterations and only until line 350 does it begin to discuss the DNA relationship. More on DNA damage and its relationship with PRM and CMA3.

Line 291.- the authors mention “sperm PRM plays such an essential role in the integrity of sperm chromatin,” this is already mentioned in line 285 “PRM dominates the sperm nucleus after replacing histones and plays a role in the final form of chromatin”.

Line 310.- the authors mention "Narrow at-the-base defects are linked to oxidative stress-induced sperm DNA damage [20]". In their research they did not measure variables to determine "oxidative stress" so this would be speculation. I recommend focusing on the findings and their interpretation.

CONCLUSION

I recommend restructuring this section and being more specific about the findings mentioned in the results.

Author Response

Dear Reviewer,

Thank you for allowing us to submit our revised manuscript titled "Sperm Head Morphology Alterations Associated with Chromatin Instability and Lack of Protamine Abundance in Frozen-Thawed Sperm of Indonesian Local Bulls" to Animals. We appreciate the time and effort you and the reviewers have dedicated to providing valuable feedback on our manuscript. We are grateful to you for your insightful comments on our manuscript.

Therefore, we use the latest line numbers to respond to your insightful comments. Here's a point-by-point response according to your suggestions on our manuscript, as follows:

Comment:

The present research provides interesting information. However, some important changes need to be made before final publication.

Abstract: review the "Journal" guidelines. It is mentioned in "MDPI Style Guide" the following: "The abstract contains a summary of the entire article and can be up to 200 words with a single paragraph." (https://www.mdpi.com/authors/layout) In this case, it exceeds the number of words. Therefore, restructure this section.

Response:

Thank you very much for your valuable suggestions and comments; we appreciate it. We have fixed it according to your suggestions.

Comment:

INTRODUCTION

General comments: I recommend adding more information on possible causes of sperm head abnormalities.

Response:

Thank you very much for your valuable suggestions and comments; we appreciate it. We have fixed it according to your suggestions.

Comment:

Line 63.- the authors mention “Pardede et al. [3] reported that sperm with a portion of total sperm abnormalities of 3-5%” estas anormalidades solo corresponden a “sperm head abnormality” ó generals?. I recommend mentioning what these anomalies would be.

Line 64.- change the word "in vivo" to "in vivo" in italics

Response:

Thank you very much for your valuable suggestions and comments; we appreciate it. We have fixed it according to your suggestions.

Comment:

MATERIALS AND METHODS

General comments: I recommend mentioning the selection criteria for these animals, body condition, etc. Also, if a complete clinical study was performed before the study to rule out any pathology.  On the other hand, why only the frozen semen was evaluated, and the semen before freezing was not considered in order to observe the pre and post-freezing effect? I recommend mentioning the number of samples evaluated for each animal.

Response:

Thank you very much for your valuable suggestions and comments; we appreciate it. We have added as much information as possible about the bulls we use. Overall the frozen semen samples used came from bulls at the AI center in Indonesia, and each bull has gone through a rigorous selection process, including clinical health issues. The sample used in the study came from local Indonesian bulls of the same breed, body weight, and age. The main reason for choosing frozen semen was because this research was conducted during the COVID-19 pandemic in Indonesia and when there were restrictions on entering the AI center directly so that only research using frozen semen could be done. In addition, pre-freezing semen samples are also impossible because it can disrupt the production process at the AI Center, which must produce a certain amount of frozen semen in a certain period. Lastly, the lack of laboratory facilities available at the AI center and the long distance between the AI center and the central laboratory makes it impossible to use semen before freezing. We've added the number of samples used on each bull.

Comment:

Line 130-131.- the authors mention "Sperm head defects in the HD group were significantly higher (P<0.05) than those in the LD group (4.03±0.13% vs. 1.91±0.13%)." I recommend moving this sentence to the "Result" section under "Characteristic sperm head morphological defects in HD and LD bulls." In addition, it would be convenient to mention that sperm with head defects were considered within the "HD" and "LD" groups as the case may be.

Response:

Thank you very much for your valuable suggestions and comments; we appreciate it. We have fixed it according to your suggestions.

Comment:

Line 113.- in the "Statistical analysis" section specify if the data came from a normal population (what tests were performed?). If they were not normal data, did you use transformed data? Also, why did you use a "Pearson correlation" and did not perform a "Sperman correlation" with the variables evaluated? Also, mention what level of significance was considered (P<0.05 or P<0.01). I suggest describing this section further.

Response:

Thank you very much for your valuable suggestions and comments; we appreciate it. The normality test on the research data was carried out using the Shapiro & Wilk test, which was then tested for homogeneity using the Levene test. The research data is normally distributed and varies homogeneously, so data analysis is continued with the T-test to compare the two groups, HD and LD. We use the "Pearson correlation" test because the data is normally distributed and homogeneous, while the "Sperman correlation" test is generally used to test correlation with abnormal data.

Comment:

RESULTS

General comments:

I recommend restructuring the figures using the same monochromatic format since Fig 3 is represented differently from Fig 4 for the "HD" and "LD" groups. As well as the labels of each axis.

Response:

Thank you very much for your valuable suggestions and comments; we appreciate it. We have fixed it according to your suggestions.

Comment:

Line 235-236.- the authors mention, "There was no difference between round head (rh), knobby acrosome (ka), detached head (dth), and abaxial (abx) in HD and LD bulls." This was already mentioned in line 228. Line 238. Below the figures is only to mention and describe the data labels. I recommend not to repeat it.

Response:

Thank you very much for your valuable suggestions and comments; we appreciate it. We have fixed it according to your suggestions.

Comment:

Line 262.- Figure 4, I recommend moving it to line 247, below the section "Sperm DNA, PRM deficiency, and PRM1 abundance in HD and LD bulls". Because there is a junction with "Table 1".

Response:

Thank you very much for your valuable suggestions and comments; we appreciate it. We have fixed it according to your suggestions.

Comment:

Line 264.- Figure 4, it mentions "DNA damage and PRM deficiency in sperm were significantly more important in HD bulls than in LD bulls (P <0.05). In addition, PRM1 concentration was significantly higher in LD bulls than in HD bulls (P <0.05)." This should go in the text, not in the figure. The text in the figure is only to describe abbreviations.

Response:

Thank you very much for your valuable suggestions and comments; we appreciate it. We have fixed it according to your suggestions.

Comment:

DISCUSSION

In general, I recommend an explanation of the findings observed in the research and try to explain the mechanism involved. I recommend restructuring this section because the first part of the discussion focuses mainly on morphological alterations, and only until line 350 does it begin to discuss the DNA relationship—more on DNA damage and its relationship with PRM and CMA3.

Response:

Thank you very much for your valuable suggestions and comments; we appreciate it. We have fixed it according to your suggestions.

Comment:

Line 291.- the authors mention, “sperm PRM plays such an essential role in the integrity of sperm chromatin,” this is already mentioned in line 285 “PRM dominates the sperm nucleus after replacing histones and plays a role in the final form of chromatin.”

Response:

Thank you very much for your valuable suggestions and comments; we appreciate it. We have fixed it according to your suggestions.

Comment:

Line 310.- the authors mention, "Narrow at-the-base defects are linked to oxidative stress-induced sperm DNA damage [20]". Their research did not measure variables to determine "oxidative stress," so this would be speculation. I recommend focusing on the findings and their interpretation.

Response:

Thank you very much for your valuable suggestions and comments; we appreciate it. We have fixed it according to your suggestions.

Comment:

CONCLUSION

I recommend restructuring this section and being more specific about the findings mentioned in the results.

Response:

Thank you very much for your valuable suggestions and comments; we appreciate it. We have fixed it according to your suggestions.

We hope the changes we have added and corrected in our manuscript will meet Animals' criteria for publication. And, of course, we hope our manuscript can contribute to animal breeding and genetics development, especially in livestock. Thank you very much again for all the suggestions and input.

Sincerely,

Authors.

Reviewer 2 Report

This study mainly aims to establish a relationship between DNA damage, PRM/ PRM1 levels and sperm defects. The study is well designed and the results present scientific soudness. Some minor corrections are suggested to improve the readibility of this work. A first sentence highlighting the main novelties in the discussion section is likely to be profitable.

L252-253, 256,259: Please add the p-value and correct r=954.

L268. This table is not reported in text. In fact this is a repetition of L250-260. I suggest to sumarize this part in the text reporting Table 1.

L274-294: These paragraphs are an “introduction” to the discussion section. I suggest to start from L295.

L370. “In our study…”

L372”As discussed in the previous paragraph …”? I suggest to rewrite both paragraphs in a sequencial order.

Author Response

Dear Reviewer,

Thank you for allowing us to submit our revised manuscript titled "Sperm Head Morphology Alterations Associated with Chromatin Instability and Lack of Protamine Abundance in Frozen-Thawed Sperm of Indonesian Local Bulls" to Animals. We appreciate the time and effort you and the reviewers have dedicated to providing valuable feedback on our manuscript. We are grateful to you for your insightful comments on our manuscript.

Therefore, we use the latest line numbers to respond to your insightful comments. Here's a point-by-point response according to your suggestions on our manuscript, as follows:

Comment:

This study mainly aims to establish a relationship between DNA damage, PRM/ PRM1 levels, and sperm defects. The study is well designed, and the results present scientific soundness. Some minor corrections are suggested to improve the readability of this work. A first sentence highlighting the main novelties in the discussion section is likely to be profitable.

Response:

Thank you very much for your valuable suggestions and comments; we appreciate it.

Comment:

L252-253, 256,259: Please add the p-value and correct r=954.

L268. This table is not reported in the text. In fact this is a repetition of L250-260. I suggest summarizing this part in the text reporting Table 1.

Response:

Thank you very much for your valuable suggestions and comments; we appreciate it. We've added the p-value and corrected r=954 according to your suggestion. We've added Table 1 links to the text. We still maintain the table and the text in our manuscript, hoping that a full description of Table 1 is contained in the text. Even so, considering that the r values are already presented in the table, we only provided the p-value for each parameter in the text.

Comment:

L274-294: These paragraphs are an "introduction" to the discussion section. I suggest to start from L295.

Response:

Thank you very much for your valuable suggestions and comments; we appreciate it. We have fixed it according to your suggestions.

Comment:

L370. "In our study…"

Response:

Thank you very much for your valuable suggestions and comments; we appreciate it. We have fixed it according to your suggestions.

Comment:

L372" "As discussed in the previous paragraph …"? I suggest to rewrite both paragraphs in sequential order.

Response:

Thank you for your valuable suggestions and comments; we appreciate it. We have fixed it according to your suggestions. Instead of rewriting the two paragraphs, we decided to improve the sentence by still adapting it to the sentences before and after without reducing the meaning of the sentence.

We hope the changes we have added and corrected in our manuscript will meet Animals' criteria for publication. And, of course, we hope our manuscript can contribute to animal breeding and genetics development, especially in livestock. Thank you very much again for all the suggestions and input.

Sincerely,

Authors.

Reviewer 3 Report

The main issue raised by the authors is the relationship between sperm head abnormalities in Indonesian bulls with DNA damage and protamine deficiencies.

The issues raised in this work are original, innovative in the field of bull andrology. The advantage of the work are also interesting results that have not been published so far. In this work, the discovery of the high usefulness of a simple and cheaper method of sperm staining in the assessment of the degree of DNA chromatin damage and the quantitative assessment of the necessary protamine deserves special emphasis. The paper contains innovative research methods useful for assessing the quality of fresh and frozen semen.

I do, however, ask the authors to clarify a few issues regarding the methodology.

When and in what season of the year were the studies conducted?

Were the bulls from one AI Center or several, where in Indonesia? Were the males in one place or in several locations? What kind of food did they have, what kind of environment, average temperature in the place where they were staying, how they were fed, what kind of food did they receive?

An interesting fact for the reader (not necessarily) may be data on local Indonesian cattle? eg body weight, what do they eat, fertility, prolificacy?

How many males were in each study group?

How many straws of frozen semen were used in total, and how many straws per bull? (important data for statistics).

For how long were semen collected from males and frozen?

line 116 -118 What exact criterion was adopted to divide the males into 2 groups?

Is it the percentage of total defects or the percentage of a single defect? What is the line that separates the two groups? What was that value or separating feature?

Was the semen also frozen for commercial purposes? Or only for research purposes? If commercial, was semen from bulls with a high percentage of head defects allowed for preservation?

How was sperm motility assessed after thawing, subjectively or objectively?

Was sperm morphology assessed in fresh semen before freezing?

Wasn't it better to assess the morphology of sperm heads in fresh semen before freezing? And divide the bulls into two groups before semen is frozen? As it is known, the freezing of sperm causes numerous damages to the acrosomes, which may indirectly affect further damage to the sperm cell nuclei after thawing the semen?

Line 98-99 of the sentence is unclear

Line 138 How many straws were used in this and subsequent studies?

Line 231 is ps in the description, shouldn't it be: pp?

table 1: PMR1-p or PRM1?

Line 321: what does KA defect mean?

Line 356: you wrote that: AO staining is the most insensitive test compared to the other tests, why did you use it?

We know the abbreviation EIA, can you give an explanation after the first mention of this abbreviation?

Author Response

Dear Reviewer,

Thank you for allowing us to submit our revised manuscript titled "Sperm Head Morphology Alterations Associated with Chromatin Instability and Lack of Protamine Abundance in Frozen-Thawed Sperm of Indonesian Local Bulls" to Animals. We appreciate the time and effort you and the reviewers have dedicated to providing valuable feedback on our manuscript. We are grateful to you for your insightful comments on our manuscript.

Therefore, we use the latest line numbers to respond to your insightful comments. Here's a point-by-point response according to your suggestions on our manuscript, as follows:

Comment:

The main issue raised by the authors is the relationship between sperm head abnormalities in Indonesian bulls with DNA damage and protamine deficiencies.

The issues raised in this work are original, innovative in the field of bull andrology. The advantage of the work are also interesting results that have not been published so far. In this work, the discovery of the high usefulness of a simple and cheaper method of sperm staining in the assessment of the degree of DNA chromatin damage and the quantitative assessment of the necessary protamine deserves special emphasis. The paper contains innovative research methods useful for assessing the quality of fresh and frozen semen. I do, however, ask the authors to clarify a few issues regarding the methodology.

Response:

Thank you very much for your valuable suggestions and comments; we appreciate it.

Comment:

When and in what season of the year were the studies conducted? Were the bulls from one AI Center or several, where in Indonesia? Were the males in one place or in several locations? What kind of food did they have, what kind of environment, average temperature in the place where they were staying, how they were fed, what kind of food did they receive?

Response:

Thank you very much for your valuable suggestions and comments; we appreciate it. The research was conducted in early 2022 during the rainy season in Indonesia. The bulls used came from the same AI center, the Singosari AI Center in East Java Province, Indonesia. All bulls are in the same location, with the same type of feed, and the environment with the same average temperature (20-22°C). Feeds are given to the bulls in the form of feed that has been mixed using the TMR (Total Mixed Ratio) method. Complete feed, or TMR, consists of forages and concentrates in balanced and adequate conditions. The feed ingredients include elephant grass, concentrate, silage, hay and minerals.

Comment:

An interesting fact for the reader (not necessarily) may be data on local Indonesian cattle? eg body weight, what do they eat, fertility, prolificacy? How many males were in each study group? How many straws of frozen semen were used in total, and how many straws per bull? (Important data for statistics). For how long were semen collected from males and frozen? line 116 -118 What exact criterion was adopted to divide the males into 2 groups? Is it the percentage of total defects or the percentage of a single defect? What is the line that separates the two groups? What was that value or separating feature?

Response:

Thank you very much for your valuable suggestions and comments; we appreciate it. The local Indonesian cattle we use are Bali bulls with a body weight of around 700 kg, with feed as described in the previous response. Regarding local Indonesian cattle fertility, we use data based on unpublished data that we have is more than 70%. Apart from Bali bulls, we have published fertility data for other local Indonesian bull breeds are Madura bulls with varying fertility rates and around 65%, lower than Bali cattle with higher fertility rates than other Indonesian local bulls. However, in this manuscript, we would like to focus on studying the relationship between sperm head damage and DNA damage and the abundance of protamine genes or proteins in sperm.

Regarding the number of bulls, we use and the criteria for dividing into groups, we have also added the line for separating bulls into two groups and the number of frozen semen samples we have used, which have been included in the latest revised manuscript. The process of collecting semen, until frozen, follows the existing AI Center protocol for around 30 hours. We did not specifically include this because our main focus is the value of sperm head defects in frozen semen. The defects listed as the criteria for grouping bulls are total defects in the sperm head and not a single defect.

Comment:

Was the semen also frozen for commercial purposes? Or only for research purposes? If commercial, was semen from bulls with a high percentage of head defects allowed for preservation? How was sperm motility assessed after thawing, subjectively or objectively? Was sperm morphology assessed in fresh semen before freezing?

Wasn't it better to assess the morphology of sperm heads in fresh semen before freezing? And divide the bulls into two groups before semen is frozen? As it is known, the freezing of sperm causes numerous damages to the acrosomes, which may indirectly affect further damage to the sperm cell nuclei after thawing the semen?

Response:

Thank you very much for your valuable suggestions and comments; we appreciate it. All frozen semen samples used were commercial frozen semen which was considered suitable for distribution based on the post-thawing motility criteria of 40%. The bulls at the AI Center had previously gone through a testing process. They were considered suitable for use as breeding bulls, including having been tested for a total abnormality of less than 20% before the freezing process. However, the overall assessment of the abnormality is not only primary defects but also secondary defects, such as tail damage. In contrast, this study focuses on primary defects, which are defects in the head due to genetic factors and greatly impact the success of sperm-oocyte fertilization. The assessment of motility after thawing is using CASA (objective test). Besides that, the main reason for choosing frozen semen was because this research was conducted during the COVID-19 pandemic in Indonesia and when there were restrictions on entering the AI center directly so that only research using frozen semen could be done. In addition, pre-freezing cement samples are also impossible because it can disrupt the production process at the AI Center, which must produce a certain amount of frozen cement in a certain period. Lastly, the lack of laboratory facilities available at the AI center and the long distance between the AI center and the central laboratory make it impossible to use cement before freezing.

Comment:

Line 98-99 of the sentence is unclear

Response:

Thank you very much for your valuable suggestions and comments; we appreciate it. We have fixed it according to your suggestions.

Comment:

Line 138 How many straws were used in this and subsequent studies?

Response:

Thank you very much for your valuable suggestions and comments; we appreciate it. We have fixed it according to your suggestions.

Comment:

Line 231 is ps in the description, shouldn't it be: pp?

Response:

Thank you very much for your valuable suggestions and comments; we appreciate it. We've fixed it to the abbreviated ps following the ones in the table. We apologize for this inaccuracy.

Comment:

table 1: PMR1-p or PRM1?

Response:

Thank you very much for your valuable suggestions and comments; we appreciate it. We have fixed it, sorry for the inaccuracy.

Comment:

Line 321: what does KA defect mean?

Response:

Thank you very much for your valuable suggestions and comments; we appreciate it. KA stands for knobbed acrosome defect. We have fixed it, sorry for the inaccuracy.

Comment:

Line 356: you wrote that: AO staining is the most insensitive test compared to the other tests, why did you use it?

Response: Thank you very much for your valuable suggestions and comments; we appreciate it. As stated in the manuscript, the statement was cited from Evenson (2016), who compared several DNA damage tests, in which we performed two tests between several DNA damages. We include this statement because of the different findings in this manuscript, where the use of Halomax appears to be more sensitive in assessing DNA damage, which can be seen from the value of DNA damage found, which tends to be higher than testing using AO. Even so, in the following sentences, we state that both tests are useful in assessing DNA damage.

Comment:

We know the abbreviation EIA, can you give an explanation after the first mention of this abbreviation?

Response:

Thank you very much for your valuable suggestions and comments; we appreciate it. We have fixed it according to your suggestions.

We hope the changes we have added and corrected in our manuscript will meet Animals' criteria for publication. And, of course, we hope our manuscript can contribute to animal breeding and genetics development, especially in livestock. Thank you very much again for all the suggestions and input.

Sincerely,

Authors.

Reviewer 4 Report

This is an interesting manuscript that aimed to analyze various alterations in the morphology of the sperm head and its association with nucleus instability and insufficient sperm protamine. In general, it is really well written and only minor revisions are suggested and listed below:

1. A few English language mistakes were found and authors are advised to provide a detailed checking for grammar.

2. In abstract, please provide the meaning for some uncommon acronyms as Acridine Orange. 

2. In material and method section, authors report that difference related to high and low sperm head defects bulls were only 4.03±0.13% vs. 1.91±0.13%. Does it presents any biological importance? Considering that semen was obtained from a central bank, how was the fertility rate for the semen donors? Additionally, please detail the statistical analysis - were data checked for normality and homocedasticity? Did you carry any variance analysis? 

3. Results are well presented and discussed. My main concern is related to the biological relevance of the data. Please include some discussions related to the number of defects found and their relevance for fertility rates. 

4. Conclusions are adequate.

I detected some minor grammar mistakes, specially in the summary and abstract. Authors should provide a detailed language revision on the manuscript. 

Author Response

Dear Reviewer,

Thank you for allowing us to submit our revised manuscript titled "Sperm Head Morphology Alterations Associated with Chromatin Instability and Lack of Protamine Abundance in Frozen-Thawed Sperm of Indonesian Local Bulls" to Animals. We appreciate the time and effort you and the reviewers have dedicated to providing valuable feedback on our manuscript. We are grateful to you for your insightful comments on our manuscript.

Therefore, we use the latest line numbers to respond to your insightful comments. Here's a point-by-point response according to your suggestions on our manuscript, as follows:

Comment:

This is an interesting manuscript that aimed to analyze various alterations in the morphology of the sperm head and its association with nucleus instability and insufficient sperm protamine. In general, it is really well written and only minor revisions are suggested and listed below:

  1. A few English language mistakes were found and authors are advised to provide a detailed checking for grammar.

Response:

Thank you very much for your valuable suggestions and comments; we appreciate it. We have fixed it according to your suggestions.

Comment:

  1. In abstract, please provide the meaning for some uncommon acronyms as Acridine Orange.

Response:

Thank you very much for your valuable suggestions and comments; we appreciate it. We have fixed it according to your suggestions.

Comment:

  1. 3. In material and method section, authors report that difference related to high and low sperm head defects bulls were only 4.03±13% vs. 1.91±0.13%. Does it present any biological importance? Considering that semen was obtained from a central bank, how was the fertility rate for the semen donors? Additionally, please detail the statistical analysis - were data checked for normality and homoscedasticity? Did you carry any variance analysis?

Response:

Thank you very much for your valuable suggestions and comments; we appreciate it. Based on the literature or previous research, it shows that the total head defects, as we reported, may not show biological importance, especially at the fertility level. However, our previous findings show that even though bulls were obtained at an AI center and have met the minimum standards for frozen semen distribution, there are still bulls that have a fertility rate of less than 55% (Pardede et al. 2022) and less than 30 % (Rosyada et al. 2020). It is possible that this was also found in the samples in this manuscript. However, we could not include data on the fertility rate of the individual bulls used. In this manuscript, we mainly only want to show that the level of damage to the sperm head in bulls is most likely due to DNA damage which may be due to the low abundance of protamine in the sperm head. However, we hope that in our next manuscript, we will be able to fully examine each of these variables, including sperm head defects and their effect on fertility rates. The normality test on the research data was carried out using the Shapiro & Wilk test, which was then tested for homogeneity using the Levene test. The research data is normally distributed and varies homogeneously, so data analysis is continued with the T-test to compare the two groups, HD and LD. We've added it to the manuscript. Thank you very much.

Comment:

  1. 4. Results are well presented and discussed. My main concern is related to the biological relevance of the data. Please include some discussions related to the number of defects found and their relevance for fertility rates.

Response:

Thank you very much for your valuable suggestions and comments; we appreciate it. We have fixed it according to your suggestions.

Comment:

  1. Conclusions are adequate.

Response:

Thank you very much for your valuable suggestions and comments; we appreciate it.

We hope the changes we have added and corrected in our manuscript will meet Animals' criteria for publication. And, of course, we hope our manuscript can contribute to animal breeding and genetics development, especially in livestock. Thank you very much again for all the suggestions and input.

Sincerely,

Authors.